# “Let’s Read Together”: A Parent-Focused Intervention on Dialogic Book Reading to Improve Early Language and Literacy Skills in Preschool Children

**DOI:** 10.3390/children9081149

**Published:** 2022-07-30

**Authors:** Raffaele Dicataldo, Meredith L. Rowe, Maja Roch

**Affiliations:** 1Department of Development and Socialization Psychology, University of Padua, 35122 Padova, Italy; raffaele.dicataldo@unipd.it; 2Harvard Graduate School of Education (HGSE), Harvard University, Cambridge, MA 02138, USA; meredith_rowe@gse.harvard.edu

**Keywords:** dialogic book reading, parent-focused intervention, preschoolers, emergent literacy, home literacy environment

## Abstract

Many children are at risk for reading difficulties because of inadequate emergent literacy skills. It is widely accepted that development of emergent literacy skills is strictly related to children’s early literacy experiences at home and school. Dialogic reading is an evidence-based intervention to promote the language skills of preschool children. This study examined the feasibility and efficacy of a parent-focused dialogic book reading intervention that aimed to foster the early language and literacy skills of pre-school children. A sample of 40 Italian preschoolers (Mage = 62.9 months, SD = 6.3) and their parents were divided into three groups: treatment group (*n* = 12); information group (*n* = 12) and control group (*n* = 16). The efficacy of the intervention for oral language skills was examined by analyzing the improvements from pre- to post-intervention in children’s oral language outcomes, through ad hoc and standardized tasks; specifically, by measuring proximal and distal abilities. Additionally, we analyzed the intervention effects on parent–child interaction and dialogic behaviors during shared book reading. Results suggest that a relatively brief intervention (6 weeks) using dialogic book reading strategies can lead to sustained improvements in early language and literacy skills in preschoolers (both proximal and distal) and in parent dialogic behaviors during shared book reading.

## 1. Introduction

Emergent literacy includes a set of interrelated linguistic skills, knowledge and attitudes identified as developmental precursors of conventional forms of reading and writing [1,2] that are pivotal for later school readiness and academic achievement [3]. Such a definition suggests that significant sources of individual differences in children’s school readiness and literacy skills are present prior to school and, therefore, need to be analyzed and faced prior to school to promote better academic outcomes [1]. It is widely accepted that emergent literacy skill development is related to children’s early literacy experiences, such as practices to which children are exposed to in kindergarten and, most importantly, at home. The home environment is a relevant and robust context for fostering oral language skills and later aspects of reading comprehension [4]. Parents are the primary source of variation in input in daily life and particularly during home literacy activities, such as storytelling, shared book reading [5] and letter naming [6]. One of the most influential home literacy parameters promoting emergent literacy development of pre-readers is the frequency of adult–child shared book reading [7]. During shared book reading, an adult reads a book to a child and uses one or more planned or structured interactive techniques to actively engage the child in the activity [8]. The fundamental role played by reading at pre-school age is evident, since it represents the linguistic, cognitive and socio-cultural foundations for children and acts both as a facilitating and protective factor for future school learning [9]. Shared book reading provides opportunities to build oral language, vocabulary and comprehension skills, as well as phonological awareness [9,10], and to focus on more narrowly constrained skills, such as print functions, directionality and book handling concepts, letter identification, concepts of words and conventions about written language [11]. It is widely agreed that variation in the frequency of shared book reading has effects on children’s language development and educational outcomes. Children who come from homes in which these activities are frequent, on average, show greater verbal precocity, develop more receptive vocabulary later, show greater knowledge of print [5,12,13,14] and, moreover, enter school with more well-developed understandings of literacy [15] compared to children from homes with fewer literacy activities. The well-known effects of shared book reading highlight the need for more research focusing on the effects of the quality of the interaction during shared book reading rather than the quantity of book reading experienced in early childhood [15,16,17]. Parents vary significantly regarding the amount of interaction they have with children during shared book reading. The amount of interaction is strongly associated with children’s trajectory of language acquisition: greater amounts of interaction and parental adjustments during interaction support children’s learning and produce a long-lasting language advantage [15,16,17,18]. The belief is that it is the interaction during shared book reading that facilitates children’s language abilities and that promotes better linguistic outcomes.

Whitehurst and colleagues [19], recognizing that greater interaction with an expert reader and more active role for children in shared book reading may be relevant to the development of children’s literacy, designed an intervention called “dialogic book reading” which aims to encourage adults to enter into dialogue with their children and interact more during shared book reading activities.

This program is based on three main principles: (a) encouraging the child to participate actively, (b) providing feedback to the child and (c) adapting adults’ reading style and language to the child’s growing linguistic abilities. Following these three principles, the adult uses a specific approach to prompting children’s participation and functions as an active listener and questioner, enabling the adult and the child to switch roles so that the child learns to become the storyteller.

Previous programs that induced caregivers to read more to children and to use instructive strategies to encourage adults to be more interactive and children to be more active reported benefits for children’s oral language, particularly in vocabulary [20,21,22,23], and broad emergent literacy skills [15,24]. Therefore, dialogic book reading is a successful activity for the development and enrichment of emerging literacy skills with a view to enhancing the linguistic, cognitive, communicative and socio-affective skills of children, who learn in a fun and enjoyable way. Evidence from the implementation of dialogic reading programs demonstrates that it is relatively easy to teach parents and early educators how to maximize shared book reading to foster language and literacy development in young children.

The main aim of this study was to analyze the feasibility and efficacy of a parent-focused intervention using dialogic book reading that aimed to foster early language and literacy skills in preschool children. In evaluating the literature on dialogic book reading programs, despite numerous studies confirming its validity, we were struck by the fact that only a few studies, notably those conducted in day cares, included a control group. Such control groups are essential because they allow for the assessment of whether children learn from regular reading sessions or whether they learn more from dialogic book reading sessions. Moreover, we found that almost all implementations of dialogic book reading to date have utilized self-instruction procedures with videotape training packages or in-person training (with or without additional videotaped explanation and examples). In addition, as reported above, the majority of research studies on dialogic book interventions used only receptive or expressive vocabulary development as a learning outcome [24,25,26]. Although shared book reading has been found to be a particularly effective technique to stimulate children’s sustained attention [27], provide a complementary source of vocabulary [28] and activate a children’s brain development [29], to date, the effects of these programs on other relevant oral language skills and, in particular, inferential skills and oral text comprehension have rarely been studied [15].

This study aimed to fill these gaps present in the dialogic book reading program literature by incorporating these important features—namely, the use of a control group and analysis of the effects of this program on other relevant oral language skills. Numerous studies have demonstrated the possibility of teaching adults the strategies of dialogic book reading, which can lead to significant changes in shared reading styles and promote the stimulation of children’s communicative and linguistic abilities [19,21,22,23,24,29]. In line with previous work, we developed a dialogic book reading program that aimed to foster the early language and literacy skills of preschoolers. In this program, fully described below, the expression “dialogic book reading” is used in a broad sense to describe an activity of reading aloud in an interactive manner that fosters the development of early language and literacy skills, such as vocabulary, sentence comprehension, inferential ability, theory of mind and narrative comprehension, as well as print-based skills.

## 2. The Current Study

The current study focused on analyzing the efficacy of an original parent-focused intervention using dialogic book reading that aimed to promote parent–child interaction during shared book reading and, in turn, foster the early language and literacy skills of pre-school children. The efficacy of the intervention for oral language skills was verified by analyzing the improvements from pre- to post-intervention in children’s oral language outcomes through ad hoc and standardized tasks; namely, by measuring proximal and distal abilities. The purpose was to address both near and far transfer effects. We tested whether the beneficial effects of storybook reading would be greater when children were active participants during shared book reading (i.e., when dialogic book reading strategies were used) as compared to when children were involved in shared book reading activities. In accordance with the research design, to assess the feasibility and efficacy of this intervention and obtain answers to specific research questions, three groups of participants were involved in this study; in detail:The treatment group (TG) represented the group who actively participated in all the intervention sessions, as described below: they received the materials, weekly assignments and support to practice with dialogic strategies;The information group (IG) represented the group who received, in conjunction with the intervention sessions for the TG, written information about language developmental milestones, the same books used during the intervention sessions and assignments to read with their children two or three times per week without receiving information and instruction about dialogic reading;The control group (CG) represented the group who received the same books used during the intervention, without any information about language development milestones or intervention.

Higher scores in all experimental tasks and on standardized tests were expected for children taking part in the intervention (TG) compared to children who participated in regular reading activities (IG and CG). Moreover, higher scores were expected for children in the IG compared to children in the CG. The decision to include a CG was driven by the desire to investigate whether the potential improvements that the children in the TG would achieve in both types of tasks (intervention-based and standardized) were due to the typical developmental trajectory and whether the amount of reading produced differences in developmental trajectories. On the other hand, the decision to include a group of parents who received only information on language development, books and weekly assignments (i.e., IG) within the research design was driven by the aim of understanding whether effects for TG children were due to the use of dialogical strategies learned directly from an expert and not by the sheer amount of reading books provided.

Moreover, we were interested in assessing the effects of this intervention on parent–child interaction during book reading. Thus, the second aim of this study was to develop an observational tool to assess parent–child interaction during book reading activities through which it would be possible to analyze whether parents learned how to use dialogic reading strategies and whether children became more participative during and after the intervention. To date, there have been few attempts to construct tools that assess the interactive reading behaviors of parents and their children, particularly in their home environment. To the best of our knowledge there is only one tool developed with this purpose; namely, the Adult–Child Interactive Reading Inventory [30]. It was not possible to use this tool since it requires extensive time and was not adaptable for evaluation in this specific program. Starting from this, we felt it was necessary to develop a new observation tool that was simple, quick to administer and tailored to our intervention—i.e., that covered all the strategies taught—thus making it useful to assess improvements in the dialogic style of both adults and children. This tool was used to code videos of interactions during shared book reading, to evaluate the efficacy of our intervention in producing behavioral change in both participants during shared book reading activities and then to analyze whether these changes mediated early language and literacy skill development after the intervention. It was expected that parents and children from the treatment group would use more dialogic strategies during shared book reading than parents and children not involved in the intervention and that the behavioral changes would explain the intervention effects on children’s outcomes.

## 3. Method

### 3.1. Procedure

This study was carried out in a preschool in the metropolitan area of (removed), a medium-sized city in northeastern Italy. In accordance with the research design, to assess the feasibility and efficacy of the intervention, as described above, three groups of participants were involved: a treatment group (TG), an information group (IG) and a control group (CG). Parents who declared their intention to be involved in the study and their availability on the dates scheduled for intervention sessions were randomly assigned to the TG or IG, whereas other participants were included in the CG. Teachers were not aware of the children’s assignments to groups or of the purpose of the intervention.

### 3.2. Participants

A group of forty children, the targets of this intervention, aged between 4 years and 4 months and 6 years and 1 month (Mage = 62.9 months, SD = 6.3) were involved in this study. Participants were recruited through direct contacts with the preschool coordinator. During the project’s presentation, organized at school, parents received information about the project, its duration and their degree of involvement in the planned activities. Children, in accordance with the differing involvement of parents in the intervention, were also divided into three groups: (1) the treatment group: *n* = 12 (F = 4; Mage = 62.6 months, SD = 6.9); (2) the information group: *n* = 12 (F = 5; Mage = 62.5 months, SD = 5.5); (3) the control group: *n* = 16 (F = 5; Mage = 63 months, SD = 6.8). Information collected through a questionnaire administered to parents showed that all children were developing typically and from families with middle socioeconomic status.

### 3.3. Materials

Children’s early oral language and literacy skills were assessed before and after the intervention with intervention-based measures, standardized tests and tasks not directly addressed in the intervention in order to investigate near and far transfer effects, respectively. Two trained master’s students individually administered all these tasks in a fixed order: none of them were aware of the children’s group assignments. The pre-testing took place over a two-week period prior to the onset of the intervention, and post-testing occurred during the week immediately following the intervention.

#### 3.3.1. Intervention-Based Measures (Near Transfer Effects)

We developed experimental probes to verify whether children benefited directly from intervention activities focused on vocabulary acquisition and comprehension of inferences.

##### Probes Targeting Vocabulary

To verify if the children learned the twenty challenging target words inserted into the storybooks used during the intervention, we developed two probes:

(1) Sentence completion task: The examiner read brief sentences describing a situation in which the target word could be inserted. The context presented in the sentences was different from the original context and from the prompt used by parents during the reading session; therefore, the children must have generalized the meaning of the new word learned to complete the sentences. Answers to each sentence were evaluated on a 0–1 point scale, where the incorrect answer was scored 0 and the correct answer 1 (range: 0–20).

(2) Word recognition task: This consisted of a set of 30 pictures depicting the 20 target words and 10 filler words. Children were asked to point out which of three pictures best represented the word orally presented by the examiner. Each item (target word) was evaluated on a 0–1 point scale, where the incorrect answer was scored 0 and the correct answer was scored 1 (range: 0–20).

##### Probes Targeting Comprehension of Inferences

To verify children’s comprehension of the two storybooks used during the intervention, we developed a task in which participants were asked if the statements about the stories were true or false. The task focused on two types of information necessary to understand the story: textual—namely, information explicitly stated in the story—and inferential—namely, information that had to be inferred. The same numbers of true and false statements were provided for each type of information. Answers to each item were evaluated on a 0–1 point scale, where the incorrect answer was scored 0 and the correct answer 1. The score consisted of the sum of correct answers, 36 for the first story and 40 for the second, with a maximum score of 76 (range: 0–76). Two separate scores, one for textual and one for inferential answers, were calculated (range: 0–38 for each).

#### 3.3.2. Standardized Tasks (Far Transfer Effects)

To evaluate the generalized effects of the intervention on early language and literacy skills, we assessed children using standardized tests and through tasks that were not directly addressed in the intervention; the tasks are described below.

Narrative comprehension (TOR 3–8): The Test for Listening Comprehension (TOR) 3–8 [31] is a standardized test for the Italian language that evaluates the narrative comprehension of children aged between 3 and 8 years of age. The tester reads two stories aloud and, to minimize the cognitive and memory load, he/she interrupts reading at two predetermined points and asks multiple-choice comprehension questions. The tester presents four alternative answers both verbally and using pictures. Participants are asked to point to the correct picture. Narrative comprehension is assessed for each story using 10 questions, half concerning information explicitly stated in the story and half requiring inferences to be generated. The score consists of the sum of correct answers, 10 for each story, with a maximum score of 20. Raw scores are transformed into standard scores with a mean = 10, SD = 2. The internal reliability, evaluated by calculating Cronbach’s alpha over items, was here 0.72.

Receptive vocabulary (PPVT-R): The PPVT-Revised [32], standardized for Italian speakers [33], is a standardized test that evaluates receptive vocabulary. It consists of a list of words, in order of increasing difficulty, presented to participants who are asked to indicate which out of four pictures best represents the target word. A basal level is defined based on the child’s ability to give eight consecutive correct answers. Testing is then continued until the participant obtains six incorrect answers out of the last eight words presented (ceiling level). Raw scores correspond to the number of correct answers minus the number of errors (range 0–175). Standard scores are computed based on raw scores: M = 100, SD = 15. The reliability for the PPVT-R, which was evaluated using the split-half procedure, was here 0.88.

Sentence comprehension (PVCL): The Test for the Evaluation of Linguistic Comprehension (PVCL) [34] is a standardized test for children aged between 3.6 and 8 years that evaluates children’s sentence comprehension. Children are presented with the form of the test appropriate for their age and required to choose which picture from among a set of four represents the sentence spoken by the experimenter. Sentences contain salient morph syntactic cues, such as gender and number agreement, conjunction, negation and different types of phrasal structures (i.e., relative, passive, temporal). One point is credited for each correct answer and the percentage of correct answers is the total raw score. Raw scores can be converted into weighted scores ranging from 0 to 100; these scores evaluate children’s overall performance considering the number of correct answers and the level of difficulty of each item.

Speeded naming (NEPSY-II subtests): The speeded naming subtest from the linguistic domain of NEPSY-II [35] was used to obtain normed measures of rapid semantic access to and production of names of colors, shapes and sizes. The child is shown an array of colors and shapes; colors, shapes and sizes; or letters and numbers and must name them in order as quickly as possible. For each item, accuracy, self-corrections and speed are recorded. Scaled scores that combine time and accuracy were here calculated (mean = 10, SD = 3). The test–retest reliability was 0.93 for the youngest children (3–4 years) and 0.72 for children between 5 and 6 years.

Inferential abilities task: An experimental task was used to assess inferential abilities. It consisted of ten items, each containing two short sentences, that were read aloud. The sentences referred to common and familiar events and were followed by two inferential questions. The questions focused on two types of inferences: knowledge-based and text-based inferences. Knowledge-based inferences required information from previously acquired world knowledge to be incorporated within the episode (e.g., “That day Piero could not wait to put on the swimsuit to play with a scoop and a bucket; Where he had gone that day?”). Text-based inferences required connecting various pieces of information provided in the short episode and identifying their implicit relations (e.g., “Then Piero picked up the scoop and the bucket. He put games in the bag; where are the scoop and the bucket?”). Answers to each question were evaluated on a 0–2 point scale: an incorrect answer was scored 0, whereas a partially correct answer or an answer provided after a clarification were scored 1 and a fully correct answer was scored 2. Three scores were calculated: total scores, consisting of the sum of correct answers, 10 for each type of inference, with a maximum of 40; knowledge-based inferences score (range: 0–20); and text-based inferences score (range: 0–20). The reliability, evaluated by calculating Cronbach’s alpha over the items, was 0.64.

Theory of mind (ToM): An experimental contents false-belief task [36] was used to assess the capacity to attribute mental states to self and others. In this task, children are shown the “pasta box” and asked to guess its content. Then, the tester shows the actual content of the box (i.e., pencils) and asks children to identify what the object is. In the control trial (“self” question), the tester asks: “What did you think was there when you saw it?” The second part of this task involves the ToM trial: another person enters the room for a while, looks at the “pasta box” and goes out; after this, the tester asks the child what the other person thought would be in the box (“other” question). Scores range between 0 and 2: either 1 (correct) or 0 (incorrect) is given for the “self” and “other” questions [37].

Print knowledge: To assess knowledge of letters and their corresponding sounds, children were shown the entire set of printed letters in random order and asked to name the letters. Total scores consisted of the sum of correct answers (0–26; number of letters included in the Italian alphabet).

#### 3.3.3. Parent–Child Interaction during Reading Activities

The Adult–Child Interactive Reading Inventory [30] is an observational tool designed to assess the joint reading behaviors of an adult and child. It contains areas for both quantitative scoring and qualitative comments. For both the adult and the child portions, the observed interactive behavior is defined by three categories: (a) enhancing attention to text, (b) promoting interactive reading and supporting comprehension and (c) using literacy strategies. Each component assesses 4 interactive behaviors for a total evaluation of 12 specific literacy behaviors.

## 4. Intervention Description

The current intervention was designed to enhance qualitative aspects of parent–child interactions shown to positively affect children’s early oral language and literacy outcomes associated with school readiness and achievement.

The intervention consisted of four educational modules implemented over six weeks and used embedded behavior-change strategies (quantitative linguistic feedback and video modeling of strategies designed to increase the quantity and quality of child-directed talk). Parents attending the program were invited to take part in four evening training sessions (2 h) held in the main room of the preschool where the children were enrolled. Each session was focused on one educational module aimed at fostering a specific linguistic skill (see Appendix A (Table A1) for an overview of the content of each module, the materials and the assignment). The four modules were built to be implemented in a sequence, with each new module building upon the content of all preceding modules.

The topics of the educational modules were as follows:Module 1: strategies and reading behaviors to increase children’s active participation

In module 1, parents were taught 15 strategies considered useful for the promotion of active participation and love of reading activities and for switching from passive reading to dialogical; i.e., responsive parenting skills that encouraged longer and strategic conversations about the book.

Module 2: strategies to foster vocabulary and to allow acquisition of new words

In module 2, parents were taught and shown the main strategies of dialogic book reading for children aged 4–5 (i.e., strategies to teach new words and foster vocabulary growth). Whitehurst and colleagues [19] developed the acronyms PEER and CROWD to help parents remember these techniques. “PEER” reminds adults to “Prompt” the child to label objects in the book and talk about the story, “Evaluate” the child’s responses, “Expand” the child’s verbalization by repeating what the child has said and adding information to it and encourage the child to “Repeat” the expanded utterances [31]. The acronym “CROWD” refers to the five types of questions that parents can formulate to engage children and encourage them to use the target word. In detail, they are: “Completion” prompts (fill-in-the-blank questions); “Recall” prompts (questions that require the child to remember aspects of the book); “Open-ended prompts” (statements that encourage the child to respond to questions); “Wh-” prompts (what, where and why questions); and “Distancing” prompts (questions that require the child to relate the content of the book to aspects of their life).

Module 3: strategies to foster inferential abilities, knowledge of story structure and narrative skills

In module 3, parents were taught and shown examples of inferences necessary to understand implicit information in books and narratives (knowledge-based and text-based). Illustrative strategies used to promote comprehension were: expanding lexical knowledge; expanding text-specific knowledge (identification of the topic); encouraging inference generation through questions; and emphasizing characteristics of the story structure. Strategies to promote narrative production were: asking about impressions about the story; asking children to recall the story (re-telling); asking questions about the main information to verify its comprehension; and asking children to add more details through specific questions.

Module 4: strategies to promote print and letter knowledge

In module 4, parents were taught and shown examples of the use of print referencing, defined as the use of any type of printed material as a tool to make a shared reading experience. In the last part of this module, how to generalize the use of the strategies learned during the training and transfer them to different materials was discussed; namely, to e-books and educational software or apps.

This intervention used a facilitator who was an expert in language development and who, in each module, through videos of parent–child interactions and ad hoc materials, illustrated dialogic shared book reading strategies useful to promote positive linguistic development outcomes. Each session had the same schedule: introduction to dialogic reading strategies with particular attention on specific language skills, presentation of short training videos on dialogic reading strategies, role-playing and games involving written instructions, discussions and delivery and demonstration of the materials used. At the end of each session, parents were provided with ad hoc materials to be used in the following week (books and instructions), a handout summarizing the strategies of dialogic book reading presented and a reminder with the assignments for the week. During the intervention, parents were asked to follow instructions, fill in weekly diaries and videotape one parent–child reading session per week. Moreover, parents were asked to send back to the facilitator, using cloud storage software, weekly diaries and parent–child reading session videos.

## 5. Results

### 5.1. Group Comparison of Intervention-Based Measures and Standardized Tasks at T1

All children completed standardized and intervention-based tasks at the two time points. Table 1 shows descriptive statistics (means, standard deviations and range) and group comparisons across the treatment (TG), information (IG) and control groups (CG) at time 1. Standard scores are reported where available.

Concerning intervention-based measures (near effects), few children were able to produce and recognize the target words, suggesting that the choice of words was accurate and that target words were distant from the vocabulary of preschool children. We found a significant difference between CG children’s performance in the word recognition task and that of children from the other two groups; namely, the IG and TG. With regard to inference comprehension, after the first plenary reading, children’s performance covered a large range of scores, with an average score of 26 out of 38 for textual information and 23 out of 38 for inferential information, showing on average good comprehension of inferences, with some weakness for comprehension of implicit information. We did not find a significant difference between children’s performances on this task.

Concerning performance in standardized tasks (far transfer effects), before the intervention, children’s average performance for receptive vocabulary (PPVT-R) was 84 (SD = 12); this is the lower boundary of the range appropriate for the age. On the other hand, performances in narrative comprehension, sentence comprehension and speeded naming were age-appropriate. Performances in the inferential abilities task covered a large range of scores, showing great variability (M = 17, SD = 6.9). Concerning letter knowledge, children demonstrated knowledge of, on average, 10 letters, whereas in the ToM task, 10% obtained scores of 0, 64% obtained 1 and 26% obtained 2. We found a significant difference between CG children’s performances in the narrative and comprehension tasks and those of children from the other two groups; namely, the IG and TG. Moreover, we found that children in the CG and TG performed better in the print knowledge task than children in the IG.

### 5.2. Intervention Efficacy for Early Language and Literacy Skills

To examine the efficacy of the intervention for early language and literacy skills, we conducted a series of mixed 3 *×* 2 ANOVAs with one between-subject factor—Group (treatment, information and control)—and one repeated-measure factor—time (pre-test, post-test). ANOVAs were conducted for intervention-based measures (proximal abilities), standardized tasks and tasks not directly addressed in the intervention (distal abilities) to analyze both near and far transfer effects.

For near effects, as it can be observed in Table 2, the ANOVAs for all intervention-based measures showed several significant group × time interactions. Bonferroni post hoc comparisons indicated that the TG had greater gains than the IG and CG for inferential answers, on both vocabulary measures, word recognition, and sentence completion task. With regard to textual answers, we found a significant difference between the TG and CG in favor of the TG and between the IG and CG in favor of the IG. Moreover, we also found a significant difference for the word recognition task between the CG and IG in favor of the IG.

Regarding far transfer effects, as it can be observed in Table 3, the ANOVAs showed significant group × time interactions for narrative comprehension and text-based inferences. Concerning narrative comprehension, the Bonferroni post hoc comparisons indicated that the TG had greater gains than the CG, whereas no differences between the TG and IG were found. With regard to text-based inferences, post hoc comparisons indicated that the TG had greater gains than the CG. In addition, Bonferroni post hoc analysis indicated that the children in the TG had greater gains than the children in the IG for the speeded naming and letter knowledge tasks.

### 5.3. Intervention Efficacy for Parent–Child Interaction

The second aim of the study was to analyze the efficacy of the intervention in increasing parent–child interaction during shared book reading. We developed an observational tool to assess and evaluate parent–child interaction during reading.

To this aim, the following steps were carried out:Step 1: Analysis of the Adult–Child Interactive Reading Inventory.

In this first step, 16 undergraduate students participating in a developmental psychology workshop were asked to assess a parent–child interaction during a shared book reading activity using the Adult–Child Interactive Reading Inventory [30], write down doubts and perplexities regarding the use of the tool and indicate any suggestions to improve the tool.

Step 2: Discussion on issues arising from the ACIRI’s use.

Starting from the evidence reported by the students, which was discussed with an expert in language development, the following changes were suggested: removing items that were not mutually exclusive; adding several new items in each category to assess children’s initiative; using a 30 s interval coding system for the categories “Promoting interactive reading and supporting comprehension” and “Using literacy strategies”; adding a global evaluation for parents’ reading style and adequacy in storytelling and for children’s involvement in reading and level of story understanding (0–5 scale); and involving two different observers for parent- and child-specific behaviors, respectively.

Step 3: Definition of the new observational tool.

The variations proposed by the students in step 2 were included in the observational tool. Moreover, a second round of coding was added that aimed to detect the frequencies of questions about temporal links and causal links and of attempts to teach words using different types of prompts (see the CROWD acronym). These changes emphasized the active role of children during shared book reading, and enabled better assessment of adult and child interaction during shared book reading; additionally, they made it possible to remove the methodological bias present in the ACIRI concerning the simultaneous observation of both participants.

The final version of the new observation tool is described in Appendix B (Table A2).

Step 4: Preliminary use of the new observational tool developed.

In order to answer the second research question, videos of interactions during shared book reading required from and sent by parents in the first and last weeks of the intervention were coded by 16 undergraduate students involved in the development of the tool. Coders were divided into couples who were randomly assigned two videos to code; moreover, each coder was asked to code, following the items included in the tool, the behavior of one of the participants of the interaction (parents or child). Coders were not aware of the groups’ assignments into dyads (TG and IG) or of the time points of the interaction; namely, pre- or post-intervention.

For this analysis, we only compared the TG and IG groups. Specifically, we coded videos of 16 dyads, of which 10 were from the TG and 6 from the IG. Considering the complexity of the observational tool, a large amount of information could have been collected; however, only a few measures, reported and briefly described in Table 4, were used for the following preliminary analysis.

We examined the equivalency of the two groups pre-test for measures collected through video coding. The results of the Mann–Whitney test indicated that the two groups were well-matched for all the above reported measures (see Table 5).

To examine whether the intervention was effective in producing changes in parent–child interaction during shared book reading after the intervention, non-parametric tests (Wilcoxon test) for each dialogic behavior detected through the observational tool were conducted pre-test (first week) and post-test (last week). As shown in Table 6, differences were found in the duration of shared book reading, in the numbers of parent questions and child answers, in the numbers of different parent and child dialogic book reading behaviors and in the total numbers of dialogic book reading behaviors used during reading. The information group did not show differences pre- and post-test; thus, we can speculate that the improvements in the dialogic behaviors of both the parents and children are attributable to our intervention.

Moreover, to preliminarily analyze the effect of the intervention on children’s linguistic outcomes—namely, vocabulary and narrative comprehension measured through standardized tests—we determined a correlation between behavioral measures at the end of the intervention and standard residuals of linguistic outcomes, thus obtaining a measure of their improvement (Table 7). Regarding children’s improvements in receptive vocabulary, a moderate correlation with the numbers of temporal and causal links promoted by parents was found (r = 0.53). Regarding children’s improvement in narrative comprehension, a significant moderate correlation with the numbers of parent dialogic book reading behaviors was found (r = 0.69), a moderate correlation with the numbers of temporal and causal link explained by children was found (r = 0.53) and a moderate correlation with the total number of parent dialogic book reading behaviors taught during the intervention was found (r = 0.72).

## 6. Discussion

Parents’ involvement in home literacy activities represents the most important source of early language input for their children [4]. Parent-focused interventions with preschoolers play a major role in the literature on disparities in early learning and development, highlighting the need to involve parents directly in the educational programs [15]. Building on a large body of research showing that shared book reading facilitates children’s language abilities [15,16,17], this study was designed to test the feasibility and efficacy of an original parent-focused intervention for dialogic reading. The intervention aimed to increase interactions between parent and child during shared book reading and, as a result, enhance the early language and literacy skills of Italian preschoolers. The efficacy of this intervention was analyzed both in terms of near transfer effects, using intervention-based tasks, and far transfer effects, using standardized tasks and tasks not directly addressed in the intervention. Additionally, as we were interested in the analysis of parent–child interaction during shared book reading, after the development of a new observational tool, we analyzed the efficacy of our intervention in producing changes in the quality of interaction; i.e., improving parent dialogic book reading behaviors that might produce changes in child behaviors and early language and literacy skills.

Concerning the first aim, although all children in the treatment, information and control groups improved their performance for each task from T1 to T2, the treatment group showed greater gains than the information and control groups in almost all intervention-based measures. After the intervention, children from the treatment group were found to know the target words and to be able to use them in different contexts, and they improved their ability to answer inferential and textual questions after they actively participated in a dialogic book reading activity. Moreover, we found that the information group obtained greater gains than the control group in the word recognition task. This means that mere repeated exposure to books, even without a specific intervention, can still lead to benefits in expanding children’s vocabulary. However, the implementation of dialogic book reading strategies produced greater results for receptive and expressive vocabulary, suggesting that the increasing parent–child interaction concerning target words during shared book reading using strategic conversations and specific prompts produced greater benefit in terms of vocabulary in general. Our results are in line with previous studies on dialogic book reading programs that demonstrated improvements in intervention-based measures of explicitly taught vocabulary, highlighting that children learn vocabulary better from conversations with adults who encourage them to talk and provide meaningful feedback on their remarks [17,18,19,20,38,39]. Moreover, we found that children in the treatment group showed great improvements in intervention-based measures of narrative comprehension, showing that dialogic book reading programs may sustain not only the improvement of receptive and expressive vocabulary but also broader linguistic skills, such as inferential abilities. Consistent with previous studies, these results highlight that the dialogic book reading program can support not only improvements in receptive and expressive vocabulary but also early literacy skills [4,5,15,40,41,42].

This evidence is also supported by the results obtained for distal abilities, which showed greater gains for the intervention group compared to the control group in narrative comprehension and text-based inferences as evaluated through the inferential abilities task. We can speculate that these results are attributable to the effects of the intervention, as one of the aspects targeted during the intervention and on which parents worked at home was promoting the generation of inferences to foster children’s narrative comprehension. It can be speculated that the intervention led to far transfer effects, providing piecemeal evidence of the generalizability of our intervention. Finally, children in the treatment group also performed better than those in the information group in the letter knowledge and speeded naming tasks. Taken together, the findings reported so far suggest that a relatively brief parent-focused intervention (6 weeks) in dialogic book reading strategies can lead to sustained improvements in early language and literacy skills in preschoolers.

Concerning the second aim, we developed a new observational tool to assess parent–child interaction during reading activities. For this aim, we compared the treatment and information groups to investigate whether changes experienced by parents and children in shared book reading activities were attributable to the participation in the dialogic book reading intervention and not to the increased amount of reading. Results from preliminary analyses showed that the duration of shared book reading activities increased more for parents in the treatment group than for parents in the information group. Moreover, for parents in the treatment group, the type and frequency of dialogic book reading behaviors increased from T1 to T2, and the number of children’s answers to parents’ questions increased. The correlation between the “dialogic behaviors” detected through the new observational tool at the end of the intervention and the improvements in receptive vocabulary and narrative comprehension suggests that this intervention had effects on parent–child interaction during shared book reading, which in turn is related to outcomes in early language and literacy skills.

The results highlight that, although the amount of shared book reading is strictly related to children’s linguistic outcomes [7], incorporation and practice of specific behaviors that increase parent–child interaction may produce greater results not only for vocabulary but also for early literacy skills related to school readiness; thus, they may also enhance future academic success for children as they enter school. Standardized tests, which are most commonly used to provide information on the progress of participants, are not useful for present purposes, as they are individual in nature and unable to demonstrate any form of interactive growth [30]. The design of a specific observational tool for this intervention provided a unique means of authenticating the progress adults and children made in their interactions during reading. Although the only observational tool aimed at evaluating parent–child interaction during shared book activities, ACIRI [30], appeared to satisfy this aim, our observational tool has more structural goals linked to the strategies specifically taught during this intervention. Moreover, in our opinion, it emphasizes the active role of children during shared book reading more and is better able to detect this role.

In conclusion, we can say that our parent-focused intervention was effective with respect to several factors. In particular, it had direct effects on the shared book reading activities carried out by parents at home. We can speculate that these changes, together with the teaching of appropriate dialogical strategies, resulted in better performances in tests related to the specific skills being trained; i.e., increased vocabulary and generation of inferences by children of parents who participated in the intervention. These findings suggest that an effective way to promote children’s literacy skills is not only to act directly on them but also to work with parents, showing them how an appropriately stimulating environment may be extremely important for their child’s emergent literacy skill development. Once parents are aware of what they can do to help their child’s development, they tend to change their behavior in a more persistent way, since they are more aware of their role as educators and of what is useful to further their child’s linguistic development [30].

The current findings highlight the central role of the family in children’s linguistic and literacy development over and above the role of the school. Although home and school are often seen as separate spheres by parents and teachers, children operate in both spaces [43]. Historically, educators thought that children’s literacy development occurs only through formal literacy instruction; nowadays, it is well-known that children gain knowledge about reading and writing through simple acts of observing and participating in informal literacy events and activities that can occur in the home environment [44]. Further studies should directly involve parents through indirect interventions to promote better oral language outcomes, in particular for children experiencing conditions of vulnerability, such as low economic status and multilingual exposure.

One of the most original contributions of the current work concerns our inclusion of an information group in addition to the classic control group. The results showed that children whose parents received only information about the stages of development of the child’s language performed better than the children whose parents received only books; namely, the control group. However, children whose parents were directly involved in the dialogic book intervention and learned strategies to interact more during the activity had better linguistic outcomes than children who were exposed with the same frequency to the shared book reading activity. These results show that, although an increase in the amount of shared book reading may produce better linguistic results, improving the quality of parent–child interaction during shared book reading may produce better, more generalized and, perhaps, longer lasting results.

However, it is important to point out, in line with previous research, how useful it is for a child’s language development to expose them to daily book reading. While dialogical reading approaches were found to produce better outcomes, improvements in children’s language skills were achieved even with simple exposure to reading, without the use of dialogical strategies, after providing parents with information about their role in the linguistic development of their children. Moreover, our findings, in line with the results of previous dialogic reading programs, show that it is relatively easy to teach parents how to maximize dialogic reading strategies that in turn promote language and literacy development in preschool children, which are strictly linked to future literacy skills [3,15,45]. Due to its potential and its easy implementation, further studies are needed to develop and validate dialogic book reading interventions on a very large scale, perhaps involving educators/teachers, who could in turn provide adequate resources for all parents to give children better opportunities to promote emergent literacy skills during early childhood.

## 7. Limitations and Future Directions

Although this study contributes to the literature on emergent literacy skill development through dialogic book reading interventions, we are aware of relevant limitations. Parents at the restitution meeting pointed out that: a 6 week intervention was not long enough to adequately address the many requests; the numbers of weekly assignments were high in relation to the time available; and some problems emerged concerning the use of home video recordings, as recording reading activities was complicated and not very spontaneous.

Another important limitation was participant dropout, which led to a significant reduction of the initial group of parents involved in the project and, therefore, meant that it was not possible to generalize the promising results of the present study. The limitations of this study should be taken into account in future studies aimed at developing and implementing dialogic book reading interventions. Our suggestion is to allow parents sufficient time to learn and implement the strategies taught, provide them with a large number of books and not be excessively demanding in order to prevent participant dropout. Moreover, in this regard, as already mentioned, it would be more effective to also involve schools and teachers in order to reach the greatest numbers of participants.

## Figures and Tables

**Table 1 children-09-01149-t001:** Characteristics of participants and group comparisons at T1.

Variables	Treatment Group (*n* = 12)			Information Group (*n* = 12)			Control Group (*n* = 16)			ANOVA (*df* = 2.37)	
	Mean	*SD*	Range	Mean	*SD*	Range	Mean	*SD*	Range	F	*p*
**Intervention-based measures**											
Sentence completion task (range: 0–20)	2.25	1.4	0–4	1.83	1.5	0–5	1.5	1.3	0–4	0.792	0.461
Word recognition task (range: 0–20)	7.25	1.6	4–10	7	2.1	4–11	5.1	2.6	2–8	4.85	0.013
Inferential questions (range: 0–38)	25.9	4.8	17–34	22.9	6.3	15–34	23.4	3.5	16–31	1.30	0.283
Textual questions (range: 0–38)	30.5	4.7	21–36	26.5	6	15–36	25.2	6.9	13–35	2.62	0.086
**Standardized tasks**											
TOR 3–8: narrative comprehension (M = 10; sd = 2)	11.9	1.7	8–15	11.1	1.5	8–14	10.1	1.3	8–12	4.74	0.022
PPVT-R: receptive vocabulary (M = 100; sd = 15)	89	7.7	75–102	86.6	13.9	68–118	79.8	12.1	65–99	2.39	0.105
PVCL: sentence comprehension (range 0–100)	66.4	14.8	43–87	61.5	13.4	39–86	47.9	17.2	20–93	5.43	0.009
Semantic access (M = 10; sd = 3)	11.1	2.2	7–14	9.0	2.9	4–12	10.1	2.6	3–15	1.89	0.165
Inferential abilities (range: 0–40)	19.8	4.1	13–26	15.1	7.3	7–28	17.5	8.1	3–31	1.36	0.269
ToM: theory of mind (range: 0–2)	1.3	0.65	0–2	1	0.60	0–2	1.1	0.61	0–2	0.877	0.425
Print knowledge (range: 0–26)	13.6	8.6	0–26	5.3	5.9	0–22	10.8	8.7	0–26	3.39	0.044

Note: SD = standard deviation.

**Table 2 children-09-01149-t002:** Means (SD) for intervention-based measures at T1 and T2 and group comparisons at T2.

Variables	Treatment Group		Information Group		Control Group		Anova Time × Group	p*ƞ**^2^*
	T1	T2	T1	T2	T1	T2		
Sentence completion	2.25 (1.4)	9.33 (4)	1.83 (1.5)	3.66 (3.3)	1.5 (1.3)	2.3 (2)	Time F(1.37) = 55.96 **	0.602
							Group F(2.37) = 13.03 **	0.413
							TxG F(2. 37) = 20.404 **	0.525
Word recognition	7.25 (1.6)	13.6 (2.3)	7 (2.1)	10.4 (3.3)	5.1 (2.6)	6.4 (2.6)	Time F(1.37) = 82.59 **	0.691
							Group F(2.37) = 18.21 **	0.496
							TxG F(2. 37) = 13.139 **	0.415
Inferential questions	25.9 (4.8)	33.5 (2.5)	22.9 (6.3)	27.1 (6.6)	23.4 (3.5)	22 (4.7)	Time F(1.37) = 31.05 **	0.456
							Group F(2.37) = 8.433 **	0.313
							TxG F(2. 37) = 19.287 **	0.510
Textual questions	30.5 (4.7)	34 (1.8)	26.5 (6)	28.5 (6.9)	25.2 (6.9)	23.4 (5.3)	Time F(1.37) = 5.988 **	0.139
							Group F(2.37) = 8.033 **	0.303
							TxG F(2. 37) = 9.086 **	0.329

Note: ** *p* < 0.001.

**Table 3 children-09-01149-t003:** Means (SD) for standardized tasks and tasks not directly addressed in the intervention at T1 and T2 and group comparisons at T2.

Variables	Treatment Group		Information Group		Control Group		Anova Time × Group	p*ƞ**^2^*
	T1	T2	T1	T2	T1	T2		
Narrative comprehension	11.9 (1.7)	12.6 (1.3)	11.1 (1.5)	10.6 (1.6)	10.1 (1.3)	11 (1.7)	Time F(1.37) = 3.050	0.076
							Group F(2.37) = 5.434 *	0.413
							TxG F(2. 37) = 4.080 **	0.227
Receptive vocabulary	89 (7.7)	97.7 (16.4)	86.6 (13.9)	94 (12.6)	79.8 (12)	87 (11.5)	Time F(1.37) = 11.583 **	0.243
							Group F(2.37) = 2.836	0.136
							TxG F(2. 37) = 0.118	0.007
Sentence comprehension	66.4 (14.8)	87.2 (8.6)	61.5 (13.4)	70.9 (18.5)	47.9 (17)	62.7 (18)	Time F(1.37) = 31.378 **	0.459
							Group F(2.37) = 9.001 **	0.327
							TxG F(2. 37) = 1.371	0.069
Speeded naming	11.1 (2.2)	12.3 (2.4)	9 (2.9)	9.3 (3.6)	10.1 (2.6)	11.5 (2)	Time F(1.37) = 5.164 *	0.121
							Group F(2.37) = 3.613 *	0.163
							TxG F(2. 37) = 0.707	0.037
Inferential abilities	19.8 (4.1)	25.1 (4.9)	15.1 (7.3)	19.3 (6.4)	17.5 (8.1)	19.2 (9)	Time F(1.37) = 21.525 **	0.389
							Group F(2.37) = 2.121	0.103
							TxG F(2. 37) = 1.843	0.091
Theory of mind	1.3 (0.65)	1.7 (0.62)	1 (0.60)	1.5 (0.67)	1.1 (0.61)	1.6 (0.61)	Time F(1.37) = 16.269 **	0.305
							Group F(2.37) = 0.978	0.050
							TxG F(2. 37) = 0.054	0.003
Print knowledge	13.6 (8.6)	16 (8.9)	5.3 (5.9)	7.4 (5.6)	10.8 (8.7)	12.1 (8.1)	Time F(1.37) = 27.743 **	0.429
							Group F(2.37) = 3.597 *	0.163
							TxG F(2. 37) = 0.782	0.041

Note: * *p* < 0.05 and ** *p* < 0.001.

**Table 4 children-09-01149-t004:** Labels and descriptions of measures collected through video coding.

Label	Definition
Length	Duration of shared book reading (in seconds)
PQuest	Number of parental questions
ChAns	Number of child answers
ParBe (range: 0–17)	Number of different parent dialogic book reading behaviors
ChBe (range: 0–17)	Number of different child dialogic book reading behaviors
ParLink	Number of temporal and causal links promoted by parents
ChLink	Number of temporal and causal links explained by children
TotPB	Total parent dialogic book reading behaviors
TotCB	Total child dialogic book reading behaviors

**Table 5 children-09-01149-t005:** Group comparisons at T1 for parent–child interaction measures.

Variables	TG (*n* = 10)		IG (*n* = 6)		Mann–Whitney Test		
	Mean	*SD*	Mean	*SD*	*U*	*z*	*p*
Length	381	133	340	163	23	−0.760	0.447
PQuest	6.2	6	9.6	9.2	23.5	−0.709	0.479
ChAns	5.1	7.8	5.6	4.5	22	−0.906	0.365
ParBe (range: 0–17)	10.6	3.4	10.1	4.8	27	−0.336	0.737
ChBe (range: 0–17)	6.8	4.4	8	2.8	22	−0.875	0.381
ParLink	2.6	3	1.5	1.9	24	−0.675	0.500
ChLink	1.2	2	0.67	1.2	30	0.000	0.991
TotPB	32	18.3	33	12.2	27	−0.326	0.745
TotCB	18.8	23.6	19.5	15.4	24.5	−0.600	0.549

**Table 6 children-09-01149-t006:** Differences between pre and post-test on parent-child interaction measures for both groups.

Variables	Treatment Group (*n* = 10)		Wilcoxon Test	Information Group (*n* = 6)		Wilcoxon Test
	T1	T2		T1	T2	
Length (in sec)	381 (133)	639 (310)	z = −2.497, *p* = 0.013	340 (163)	357 (131)	z = −0.734, *p* = 0.463
PQuest	6.2 (6)	20.6 (13)	z = −2.666, *p* = 0.008	9.6 (9.2)	9.8(18.2)	z = −0.405, *p* = 0.686
ChAns	5.1 (7.8)	14.6 (9.6)	z = −2.253, *p* = 0.024	5.6 (4.5)	5.5 (7.5)	z = −0.677, *p* = 0.498
ParBe (range 0–17)	10.6 (3.4)	13.5 (2.2)	z = −2.199, *p* = 0.028	10.1 (4.8)	9.1 (3)	z = −0.946, *p* = 0.344
ChBe (range 0–17)	6.8 (4.4)	12.4 (4.6)	z = −2.706, *p* = 0.007	8 (2.8)	9.5 (4.5)	z = −0.420, *p* = 0.674
ParLink	2.6 (3)	4 (4.6)	z = −1.292, *p* = 0.196	1.5 (1.9)	1.6 (3.1)	z = 0.000, *p* = 1
ChLink	1.2 (2)	2.4 (2.5)	z = −1.794, *p* = 0.073	0.67 (1.2)	0.50 (1.2)	z = −0.272, *p* = 0.785
TotPB	32 (18.3)	51.4 (22)	z = −2.710, *p* = 0.007	33 (12.2)	28.1 (17.3)	z = −0.734, *p* = 0.463
TotCB	18.8 (23.6)	34.4 (16)	z = −2.091, *p* = 0.037	19.5 (15.4)	25.1 (19.9)	z = −0.736, *p* = 0.462

**Table 7 children-09-01149-t007:** Correlations between parent-child interaction measures and improvement in receptive vocabulary and narrative comprehension.

	1	2	3	4	5	6	7	8	9	10	11
Improvement in Vocabulary	-	−0.07	−0.19	−0.34	−0.43	0.44	0.14	0.53 *	0.17	−0.01	−0.32
Improvement in Narrative comprehension		-	0.66 **	0.42	0.55 *	0.69 **	0.34	0.18	0.53 *	0.71 **	0.57 *
Length			-	0.44	0.53 *	0.38	0.43	0.19	0.62 *	0.81 **	0.66 **
Number Parent Questions				-	0.91 **	0.10	0.27	−0.05	−0.01	0.46	0.54 *
Number Child Answers					-	0.18	0.28	−0.13	0.04	0.50	0.60 *
Number different Parent dialogic behaviors						-	0.49	0.58 *	0.55 *	0.70 **	0.38
Number different Child dialogic behaviors							-	0.39	0.72 **	0.55 *	0.75 **
Number Parent temporal and causal link								-	0.45	0.28	0−.01
Number Child temporal and causal link									-	0.66 **	0.57 *
Total Parent dialogic behaviors										-	0.72 **
Total Child dialogic behaviors											-

Note: * *p* < 0.05; ** *p* < 0.001.

## Data Availability

The datasets used during the current study are available from the corresponding author on reasonable request.

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
