# Peer review of "“Let’s Read Together”: A Parent-Focused Intervention on Dialogic Book Reading to Improve Early Language and Literacy Skills in Preschool Children"

_children, 2022, doi:10.3390/children9081149_

Round 1
Reviewer 1 Report
“Let’s read together” a Parent-focused intervention on dialogic book reading to improve early language and literacy skills in preschool children
The present study examined the efficacy of a parent-focused intervention on shared book reading. It is interesting that the study examined both: the improvements in children on oral language skills, as well as, the effects on parent-child interaction during shared book reading. The weakest parts of the article are: 1. References are outdated (Most of them pre- 2000). 2. The baseline does not seem to be the same for the participant groups at least in the standardized tasks.
Introduction
Authors presented three terms indistinctly along the Introduction: reading book aloud (line 48), shared reading (line 59) and dialogic book reading (line 75). It would be convenient to briefly explaining the differences between them.
Lines 63-64: “quality of the interaction during shared book reading rather than quantity…” This statement is supported by only one reference published in 1985. You should review more recent references on this topic before making such assertion: for example, Bojczyk, Davis & Rana (2016): Mother-child interaction quality in shared book reading….; Walsh, (2008): Quantity, Quality, Children’s Characteristics and Vocabulary Learning; Kassow (2006) Parent-child shared book reading: quality versus quantity…. among others.
Lines 100-105: “Moreover, we found that almost all implementations of dialogic reading to date…. … In addition, as reported above…. Book interventions frequently used only receptive or expressive vocabulary” Please, you should see more recent references on this topic before making such a statement. In fact, share book reading has been considered a particularly effective technique for scaffolding young child’s sustained attention (Lawson, 2012), providing a complementary source of vocabulary (Montag, Jones, & Smith, 2015), activating child’s brain development (Hutton et al., 2017) etc… The list of recent studies on this topic continues…
Line 108 “these important features”. What features do you refer to? Please specify.
Line 128: What do you mean by a regular book-reading situation?
Lines 159-160: “To date there have been few attempts…” Please you should review recent literature on this topic.
Method
The characteristics of participants at T1 stablished differences among them for example it seems that the TG has significant higher abilities in the word recognition, task the narrative comprehension, the sentence comprehension than the CG. The baseline does not seem to be the same for both groups.
What is the goal to use a ToM test in this study?
It would be appreciated if the intervention description, the intervention efficacy on parent-child
will be presented in a more synthetic way, highlighting only the most important contributions.
Lines 144-145 “this is the first study 144 on dialogic book-reading with preschoolers that uses early language and literacy skills and not only receptive or expressive vocabulary as learning outcomes” I do not think that this is the first study on this topic. Please review (among others):
1. Hindman, Connor, Jewkes & Morrison (2008) Untangling the effect of shared book reading…
2. Phillips, L. M., Norris, S. P., & Anderson, J. (2008). Unlocking the door: Is parents' reading to children the key to early literacy development? Canadian Psychology/Psychologie canadienne, 49(2), 82–88. https://doi.org/10.1037/0708-5591.49.2.82
3. Aram, D. & Shapira, T. (2012). Parent-Child shared book reading and Children’s language, literacy, and empathy development
4. Sim, S. & Berthelsen, D. (2014). Shared Book Reading by Parents with Young Children: Evidence-Based Practice
5. Hindman, AH., Skibbe, LE, Foster, TD. (2014). Exploring the variety of parental talk during shared book reading and its contributions to preschool language and literacy: Evidence from the Early Childhood…
6. Deckner, DF., Adamson, LB., Bakeman, R. (2006). Child and maternal contributions to shared reading: Effects on language and literacy development
References
References are outdated (Most of them pre- 2000). There are a large number of studies on this subject published after 2000, you should make an exhaustive search of the databases on this subject and support your study with more recent references.
Author Response
Author's Reply to the Review Report
Answers to Reviewer 1:
“Let’s read together” a Parent-focused intervention on dialogic book reading to improve early language and literacy skills in preschool children
The present study examined the efficacy of a parent-focused intervention on shared book reading. It is interesting that the study examined both: the improvements in children on oral language skills, as well as, the effects on parent-child interaction during shared book reading. The weakest parts of the article are: 1. References are outdated (Most of them pre- 2000). 2. The baseline does not seem to be the same for the participant groups at least in the standardized tasks.
We would like to thank reviewer 1 for the accurate and valuable comments that help us improve our paper. We have tried to respond to all of them. Below you can see how we have addressed each point. We hope that this has resulted in an overall improvement of our work.
Introduction
Authors presented three terms indistinctly along the Introduction: reading book aloud (line 48), shared reading (line 59) and dialogic book reading (line 75). It would be convenient to briefly explaining the differences between them.
We are sorry for the lack of clarity regarding the use of these terms. Following reviewer 1's suggestion, we have included the definitions of shared reading and dialogic book reading, for greater understanding.
Shared reading: “During shared book reading, an adult reads a book to a child or a group of children and uses one or more planned or structured interactive techniques to actively engage the children in the text”;
Dialogic book reading: “… the adult uses a specific approach to prompting children’s participation and functions as an active listener and questioner, enabling the adult and the child to switch roles so that the child learns to become the storyteller”.
Lines 63-64: “quality of the interaction during shared book reading rather than quantity…” This statement is supported by only one reference published in 1985. You should review more recent references on this topic before making such assertion: for example, Bojczyk, Davis & Rana (2016): Mother-child interaction quality in shared book reading….; Walsh, (2008): Quantity, Quality, Children’s Characteristics and Vocabulary Learning; Kassow (2006) Parent-child shared book reading: quality versus quantity…. among others.
We thank reviewer 1 for pointing out some more recent literature references. Although our literature review was thorough, the suggested literature had been left out in our search. Therefore, following his/her suggestion, we searched for and cited more recent literature references.
Lines 100-105: “Moreover, we found that almost all implementations of dialogic reading to date…. … In addition, as reported above…. Book interventions frequently used only receptive or expressive vocabulary” Please, you should see more recent references on this topic before making such a statement. In fact, share book reading has been considered a particularly effective technique for scaffolding young child’s sustained attention (Lawson, 2012), providing a complementary source of vocabulary (Montag, Jones, & Smith, 2015), activating child’s brain development (Hutton et al., 2017) etc… The list of recent studies on this topic continues…
We thank reviewer 1 for pointing out some more recent literature. We tried to explain better what was meant and added some literature references related to the effect of intervention programs using dialogic reading strategies on domains other than language. “Although, shared book reading has been found to be a particularly effective technique to stimulate children's sustained attention, provide a complementary source of vocabulary, activate a child's brain development, to date the effects of these programs on other relevant oral language skills and in particular inferential skills and oral text comprehension have rarely been studied”
Line 108 “these important features”. What features do you refer to? Please specify.
Following this suggestion, we rewrote: “This study aims to fill these gaps present in the dialogic book reading program literature, incorporating these important features, namely, the use of a control group and analysis of the effects of this program on other relevant oral language skills”.
Line 128: What do you mean by a regular book-reading situation?
What was meant by this sentence was that we would compare who during the shared readings took advantage of the strategies of dialogic reading and who did not (mere shared reading). To make understanding easier, we rewrote: “We tested whether the beneficial effects of storybook reading would be greater when children were active participants during shared book reading (i.e. when dialogic book reading strategies are used) as compared to when children were involved in a shared book-reading activities”.
Lines 159-160: “To date there have been few attempts…” Please you should review recent literature on this topic.
As reported in this paper, to the best of our knowledge there is only one instrument developed with the aim to assess parents and children's interactive reading behaviors, namely the Adult-Child Interactive Reading Inventory. We have not found other tools with this aim.
Method
The characteristics of participants at T1 stablished differences among them for example it seems that the TG has significant higher abilities in the word recognition, task the narrative comprehension, the sentence comprehension than the CG. The baseline does not seem to be the same for both groups.
We are aware that the baseline was not the same for the groups, however since this was a pilot study we used a convenience group and could not control the allocation of participants in the research design.
What is the goal to use a ToM test in this study?
Since some of the inferences enhanced during the dialogic reading activities included in the intervention focused on internal states and intentions of the characters, we found it useful to observe any effects on theory-of-mind skills as well.
It would be appreciated if the intervention description, the intervention efficacy on parent-child will be presented in a more synthetic way, highlighting only the most important contributions.
Following your suggestion, we reduced the description of the intervention by removing redundant information. In addition, we have reduced the description of the methodology used to construct the instrument and partly the results section. We hope that these changes make the paper easier to read.
Lines 144-145 “this is the first study 144 on dialogic book-reading with preschoolers that uses early language and literacy skills and not only receptive or expressive vocabulary as learning outcomes” I do not think that this is the first study on this topic. Please review (among others):
- Hindman, Connor, Jewkes & Morrison (2008) Untangling the effect of shared book reading…
- Phillips, L. M., Norris, S. P., & Anderson, J. (2008). Unlocking the door: Is parents' reading to children the key to early literacy development? Canadian Psychology/Psychologie canadienne, 49(2), 82–88. https://doi.org/10.1037/0708-5591.49.2.82
- Aram, D. & Shapira, T. (2012). Parent-Child shared book reading and Children’s language, literacy, and empathy development
- Sim, S. & Berthelsen, D. (2014). Shared Book Reading by Parents with Young Children: Evidence-Based Practice
- Hindman, AH., Skibbe, LE, Foster, TD. (2014). Exploring the variety of parental talk during shared book reading and its contributions to preschool language and literacy: Evidence from the Early Childhood…
- Deckner, DF., Adamson, LB., Bakeman, R. (2006). Child and maternal contributions to shared reading: Effects on language and literacy development
We thank reviewer 1 for pointing out more recent literature. Accordingly, we rewrote the sentence: “Consistent with previous studies, these results highlight that the dialogic book-reading program can support not only improvements in receptive and expressive vocabulary, but also early literacy skills”. Additionally, based on the bibliographical references indicated by reviewer 1, we have partly modified introduction and discussion.
References
References are outdated (Most of them pre- 2000). There are a large number of studies on this subject published after 2000, you should make an exhaustive search of the databases on this subject and support your study with more recent references.
We thank reviewer 1 for pointing out some more recent literature references that we have incorporated into this version of the paper.

Reviewer 2 Report
The authors are describing and answering a complex set of problems in an intervention-based approach to home reading practices to better promote literacy outcomes for early readers. I think this is important work that may ultimately lead to changes in home literacy practices for families and children. My only concern is that the many typos, run-on sentences, and overly wordy writing style may be distracting and confusing, thus undermining the impact of the study. I recommend a careful read-through for typos, as well as removal of unnecessary turns of phrase particularly in the results and discussion.
More specific comments:
Participants- I would like a lot more information about the socio-economic status and educational backgrounds of the families. What exactly defines “middle socioeconomic status”?
Results- The opening paragraph of the results section is unnecessary.
Results- Consider including a graph showing treatment group and performance over time. Such an illustration would make the results more impactful.
Discussion- I'd like a reflection on the impact of family socioeconomics on home reading practices and the degree to which these intervention practices may be out of reach for certain low-income communities. I am also interested in the authors' thoughts on how to implement interventions.
Author Response
Author's Reply to the Review Report
Reviewer 2:
The authors are describing and answering a complex set of problems in an intervention-based approach to home reading practices to better promote literacy outcomes for early readers. I think this is important work that may ultimately lead to changes in home literacy practices for families and children. My only concern is that the many typos, run-on sentences, and overly wordy writing style may be distracting and confusing, thus undermining the impact of the study. I recommend a careful read-through for typos, as well as removal of unnecessary turns of phrase particularly in the results and discussion.
We would like to thank reviewer 2 for the accurate and valuable comments that help us improve our paper. Following the suggestions of both reviewers, we have reduced part of the description of the intervention, the part related to the construction of the parent-child interaction assessment tool, summarized part of the results and reduced the discussion section. Finally, we did a careful rereading and eliminated typos and run-on sentences, to make the paper read more smoothly.
More specific comments:
Participants- I would like a lot more information about the socio-economic status and educational backgrounds of the families. What exactly defines “middle socioeconomic status”?
We measured SES as the education level of both parents and the annual family income level. In both cases, data were collected categorically: parental educational levels were classified into 6 categories (1 = primary education degree, 2 = middle school degree, 3 = high school degree, 4 = bachelor’s degree, 5 = master’s degree, and 6 = post-graduate education) and annual income was coded based on a 5-point scale (1 = below EUR 24,000, 2 = EUR 25,000–30,000, 3 = EUR 31,000–34,000, 4 = EUR 35,000–40,000, 5 = above EUR 40,000). Categories were transformed into a continuous scale of years of education and Euro (thousands).
We defined “middle socioeconomic status” those families in which parents have middle school degree and the annual income ranging from 30-34 thousand, equivalent to the Italian median family income.
Results- The opening paragraph of the results section is unnecessary.
We removed this opening paragraph
Results- Consider including a graph showing treatment group and performance over time. Such an illustration would make the results more impactful.
Although we agree with reviewer 2 that a graph would make the results more impactful, we find it difficult to fit all the trajectories in the different skills assessed into one graph so we feel that tables meet our needs.
Discussion- I'd like a reflection on the impact of family socioeconomics on home reading practices and the degree to which these intervention practices may be out of reach for certain low-income communities. I am also interested in the authors' thoughts on how to implement interventions.
An important mechanism by which SES operates on children’s language development is the literacy-related resources and interactions that young children experience at home, especially during the preschool years. Family SES may be considered as one of those static factors that influence the features of the Home Literacy Environment for preschoolers. Family income affects the availability of material resources that can influence the quantity of input and thereby child development, whereas parental educational history is linked to non-material resources, including the parent’s academic competence, attitudes toward education, knowledge and beliefs about child development, and overt behavior. Low SES seems to be a risk factor for HLE, however, recent studies shown that there are also variations in the quality of HLE among low SES families. Despite many economic burdens, many low SES families put a great effort into supporting literacy, though, for example, shared book reading.
Usually, intervention practices may be out of reach for low-income families. There can be many reasons for this, some related to financial constraints that limit the use of interventions that involve the purchase of materials but also just the need to attend classes and thus having to organize with work and family management. Others may relate to gaps in knowledge of child development and beliefs about educating one's children, delegating it to the school system. However, there are families who although they have low incomes, have a focus on the development of their children and by turning to family support services or associations in the area are able to access these kinds of initiatives. Family support services are particularly important for parents with limited access to material resources, with limited support from extended family and those who are socially isolated. Family services can also help families cope with personal or family circumstances that affect parental engagement and the quality of time they spent with their children. A crucial element to effectively combat poverty and its effects is to provide services that meet the needs of children and parents, to prevent and/or repair the possible consequences of poverty on children's well-being and development. For instance, family services play a crucial role in improving children’s material living environment, to reduce parental stress and create a supportive home learning environment. In this way, by promoting these kinds of initiatives at family support services or associations well established in the area, we believe that it is possible to reach and involve parents and children from families of low-SES.
